# Survival and Enrichment Analysis of Epithelial–Mesenchymal Transition Genes in Bladder Urothelial Carcinoma

**DOI:** 10.3390/genes14101899

**Published:** 2023-09-30

**Authors:** Waleed Ali, Weirui Xiao, Daniel Jacobs, Andre Kajdacsy-Balla

**Affiliations:** 1Albert Einstein College of Medicine, New York, NY 10461, USA; weirui.xiao@einsteinmed.edu (W.X.); daniel.jacobs2@einsteinmed.edu (D.J.); 2Professor of Pathology, University of Illinois at Chicago College of Medicine, Chicago, IL 60607, USA; aballa@uic.edu

**Keywords:** epithelial–mesenchymal transition, bladder urothelial carcinoma, TCGA, VEGF, in silico

## Abstract

The escalating prevalence of bladder cancer, particularly urothelial carcinoma, necessitates innovative approaches for prognosis and therapy. This study delves into the significance of genes related to epithelial–mesenchymal transition (EMT), a process inherently linked to carcinogenesis and comparatively better studied in other cancers. We examined 1184 EMT-related gene expression levels in bladder urothelial cancer cases through the TCGA dataset. Genes shown to be differentially expressed in relation to survival underwent further network and enrichment analysis to uncover how they might shape disease outcomes. Our in silico analysis revealed a subset of 32 genes, including those significantly represented in biological pathways such as VEGF signaling and bacterium response. In addition, these genes interact with genes involved in the JAK-STAT signaling pathway. Additionally, some of those 32 genes have been linked to immunomodulators such as chemokines CCL15 and CCL18, as well as to various immune cell infiltrates. Our findings highlight the prognostic utility of various EMT-related genes and identify possible modulators of their effect on survival, allowing for further targeted wet lab research and possible therapeutic intervention.

## 1. Introduction

Bladder cancer is the 10th most common malignancy worldwide with 573,278 new cases and 212,536 deaths in 2020 [1]. Urothelial carcinoma accounts for over 90% of bladder cancers, which costs the U.S. alone $4 billion annually [2]. The prevalence is predicted to continue to rise due to the increasing industrialization and urbanization in developing countries, and the aging population [3,4]. Well-studied risk factors include cigarette smoking and occupational exposures especially in metal workers, painters, and chemical process workers [5,6]. Various altered genes have been implicated in amplifying the effect of these environmental exposures including carcinogen detoxification genes like *UDP Glucuronosyltransferase Family 1 Member A Complex Locus* (*UGT1A*) and *N-acetyltransferase 2* (*NAT2*) [7,8,9], fibroblast growth factor receptor 3 (*FGFR 3*) [10], *p16*, *p53*, retinoblastoma (*RB*), matrix metalloproteinases, genes involved in folate metabolism, and high activity metabolic activators like high activity P450 cytochrome enzymes [11,12,13,14,15,16,17]. Within the TNM staging used, T1 represents tumor invasion up to the muscular layer of the bladder, with further stages T2,3,4 representing further progression into the muscle, perivesicular layer, and adjacent structures and organs, respectively. Currently, treatment options include transurethral resection of bladder tumor (TURBT) and intravesical therapy for non-muscle invasive, while options for metastatic disease include radial cystectomy, neoadjuvant chemotherapy, and newer immunotherapies [2,18,19,20,21,22].

Several genes have been implicated in the development of urothelial cancer and metastases and used as targets for therapy. Some of these include *LRP1B* [23], *ERRC2*, *FANCC*, *ATM*, *RB1* [24], *p53* [25], and *SLC14A1* [26]. There are currently no widely accepted bladder cancer screening programs, though biannual cystoscopies have been found to be efficacious in vulnerable subpopulations [27,28,29]. Currently follow-up is time-consuming and expensive, consisting of cystoscopy, imaging, and surgery; urine biomarkers are being studied to supplement those options [18,19].

Cells involved in the invasion of bladder cancer alter their surroundings and can also become transiently and reversibly plastic, turning into mesenchymal stem cells. This is the epithelial–mesenchymal transition (EMT), which is among the most relevant paradigm shifts in how we view cancer progression and can combat its growth. During EMT, epithelial cells lose their polarized, adhesive characteristics and gain a mesenchymal phenotype, enabling them to migrate and invade surrounding tissues [30,31]. Transcription factors including Snail, Zeb, and Twist aid in this process by repressing E-cadherin, an epithelial transmembrane protein [32]. In contrast to epithelial cells, mesenchymal carcinoma cells exhibit specific metabolic needs. As they undergo EMT, cancer cells finely regulate multiple metabolic pathways to support the demands of rapid cell proliferation [33]. The molecular pathways shown to be associated with EMT include Snail/Slug, Twist, Six1, Cripto, TGF-β, and Wnt/β-catenin [34]. The literature shows how genes such as *CDH1*, *ZEB1*, *TGFB*, *CDH2*, *VIM*, and *TIMP1* have been linked to inducing the EMT phenotype, driving cell migration, and adapting to changing demands on the primary tumor [33,35,36].

In bladder cancer, various microRNAs (miRNAs) have been found to regulate proteins such as Smad7 or Twist1, either promoting or disrupting EMT and metastasis [37]. Understanding the mechanisms underlying EMT is crucial for developing targeted therapies to control cancer metastasis and may prove useful in treatment options going forward. Comparatively, there has been less work in this field in bladder cancer than in other cancers. This paper examines a multitude of EMT-related genes in relation to not only outcomes, but also the biologic networks and pathways which allow these genes to influence carcinogenesis and affect these outcomes.

## 2. Materials and Methods

### 2.1. Selection of Genes

To have a comprehensive overview of genes involved in the epithelial–mesenchymal transition, dbEMT 2.0 (http://dbemt.bioinfo-minzhao.org/ (accessed on 1 September 2022)), a database curated for focus on EMT-related genes, was utilized. A spreadsheet was generated with 1184 genes listed on the database, obtained from an initial PubMed abstract query for “Epithelial Mesenchymal Transition Genes” with the results mined for unique genes linked to EMT (see Appendix A).

### 2.2. Survival Analysis

Publicly available cases from the NIH-funded “The Cancer Genome Atlas” (TCGA) project were utilized to examine gene expression pertaining to survival in bladder urothelial cancer (data portal: https://portal.gdc.cancer.gov/projects/TCGA-BLCA (accessed on 1 September 2022)). Kaplan–Meier plots were generated through the R2 platform (https://hgserver1.amc.nl/cgi-bin/r2/main.cgi (accessed on 1 September 2022)) using the TCGA dataset for “Bladder Urothelial Carcinoma”, *n* = 407. The built-in “KaplanScan” algorithm was used to divide mRNA gene expression into “high” versus “low” categories (*n* values for each based on KaplanScan groupings of expression). Overall survival was compared to follow-up time in months being analyzed. For multiple hypothesis testing, *p*-values were adjusted to a false discovery rate (FDR) of 0.05.

### 2.3. Expression Analysis

To compare normal versus tumor levels of those EMT genes which showed differential expression regarding survival, mRNA levels for the “Bladder Urothelial Carcinoma” TCGA dataset were analyzed with a Welch’s *t*-test through the UCSC Xena platform. This platform is a genome browser and visualization tool of genomic and phenotypic data for both public and private datasets (https://xena.ucsc.edu/ (accessed on 1 September 2022)). An FDR cutoff of 0.05 was used for significance. Violin plots were generated to visualize expression values.

Through the aforementioned R2 platform, the expression levels of the previously mentioned genes which showed differential expression in regard to survival were analyzed. mRNA gene levels were compared in respect to pathologic staging and, given the small *n* associated with cases of stage 1 bladder urothelial cancer in the TCGA dataset (*n* = 2), Kruskal–Wallis analysis with corresponding pairwise Welch’s *t*-tests were used. Again, *p*-values were adjusted and an FDR cutoff of 0.05 was used.

### 2.4. Network & Enrichment Analysis

GeneMANIA (http://www.genemania.org (accessed on 1 September 2022)) serves as a platform for visualizing diverse biological interactions encompassing co-expression, co-localization, and domain similarity. In this study, GeneMANIA, R (https://www.r-project.org/ (accessed on 1 September 2022)), and the Cytoscape platform (https://cytoscape.org/ (accessed on 1 September 2022)) were utilized to construct a gene–gene interaction network focusing on EMT-related genes exhibiting significant differential expression based on survival data.

Subsequently, the network analysis highlighted certain genes alongside the aforementioned EMT genes. These genes underwent enrichment analysis to shed light on their potential involvement in specific biological processes, using annotations from the Gene Ontology (GO), Kyoto Encyclopedia of Genes, and Genomes (KEGG) databases. It was carried out through the Metascape platform (http://metascape.org (accessed on 1 October 2022)). The criteria for this analysis included a minimum overlap of 3 genes and an enrichment threshold of 1.5. Statistical significance was set at *p* < 0.05.

### 2.5. Tumor Immune Microenvironment Analysis

The relationship between the EMT-related genes differentially expressed in connection with survival and the immune system in cases of bladder urothelial cancer was examined through the TISIDB and TIMER platform. Spearman’s correlations between mRNA gene expression of the EMT genes and clinically relevant immunoinhibitor and cytokine gene expressions were calculated and visualized. Those with a rho of >|0.40| were considered meaningful with a *p*-value of >0.01. Devolution methods were used to estimate the immune infiltration of a wide variety of immune cells based on gene expression on the TIMER platform. The Spearman’s correlation was adjusted based on tumor purity, with a rho of >|0.30| visualized.

## 3. Results

### 3.1. Survival Analysis

A myriad of EMT-related genes showed differential expression in normal versus cancer tissues. Kaplan–Meier plots (Figure 1) were generated using TCGA data on “Bladder Urothelial Carcinoma” for each of the 1184 genes mined from the EMTdb, with only 32 meeting significant cutoff following FDR correction (Table 1).

Genes with “high” mRNA expression leading to worse prognosis include those in Table 2.

On the other hand, genes which showed “low” mRNA expression leading to worse prognosis include those in Table 3.

### 3.2. Expression Analysis

Out of the 32 genes highlighted in the survival analysis, some of them also showed statistically significant expression levels when comparing normal versus tumor samples. This can be seen in the density plot (Figure 2) or in more detail in the sample violin plots (Figure 3); the rest of the genes and *p*-values from the violin plots in Appendix A can be seen in Table 4. Some genes also showed significant differential expression based on tumor stage (Figure 4).

From the XENA browser, 19 normal tissues were compared with 407 TCGA cases. Following FDR correction (cut-off *p*-value is 0.003143), 10 genes were shown to be differentially expressed when considering overall survival (Table 5).

Several genes approach significance post-correction (Table 6).

Out of the same set of EMT-genes regarding survival, 10 out of the 32 showed differential expression when considering pathologic staging (Table 7).

### 3.3. Identification of Further Gene Interactions and Enriched Biological Processes

#### 3.3.1. Network Analysis

Network analysis was employed to identify genes with interconnected relationships, utilizing factors such as physical interactions, co-localization, and co-expression data (Figure 5). The genes chosen for constructing each network analysis were the previously mentioned EMT-associated genes that displayed distinct expression patterns in survival outcomes.

#### 3.3.2. Enrichment Analysis

Enrichment analysis was executed on two distinct gene sets: first, on the genes pinpointed in the network analysis, and additionally, on the genes situated at the periphery (“outer rim”) of the network analysis, apart from the initial EMT genes (Figure 6). Enrichment analysis of the initial 32 highlighted EMT genes showed numerous pathways significantly overrepresented in the cohort of genes, such as endothelial cell migration and regulation of cell shape, in line with the known physiologic processes involved with the epithelial–mesenchymal transition. However, other less canonically associated ones such as defense response to bacterium and carbohydrate response were also uncovered through the analysis. Regarding the genes listed in the network analysis, similar biologic processes such as the VEGFA signaling pathway was highlighted alongside other related ones such as HIF-1 survival signaling.

### 3.4. Correlation to Inflammation Mediators

#### 3.4.1. Immunomodulator, Cytokine

Through examining for correlation between various immunomodulators and chemokines (see Appendix B, Table A1 for list), the following plots were generated (Figure 7), with the statistically and clinically significant ones (*p* < 0.05 post correction, |rho| > 0.4) being shown. Out of the 32 genes, 12 were shown to have at least significant correlation with an immunomodulator or chemokines (TBX3, NRP2, FN1, FOXA1, FBP1, ANXA1, LAMC2, HOOK1, NES, PTPN6, RUNX2), with the first four having at least 25 different immunomodulators or cytokines to be significantly correlated with (Table 8).

#### 3.4.2. Immune Cell Infiltrate

Through the TIMER platform, deconvolution methods were used to estimate the amount of various immune cells (T cells CD4+, Tregs, B cells, Neutrophils, Monocytes, Macrophages, DCs, NK and Mast Cells; see Figure 8). Out of the 32 initial genes, 14 (ADAM17, AGER, ANXA1, ARMC8, FBP1, FN1, FOXA1, LAMC2, MAP2K1, NRP2, PTPN6, RUNX2, STIM2, TBX3) showed to have at least significant correlation (Spearman’s correlation > 0.05 following adjustment based on tumor purity, with a rho of >|0.30|; see Table 9).

## 4. Discussion

Overall, we see a robust cohort of EMT-related genes that are differentially expressed as pertaining to survival in bladder urothelial carcinoma. *RUNX2*, a gene most associated with cartilage production, has been linked to pancreatic cancer and to breast cancer progression through modulation of MicroRNAs and the metastasis-associated 1 (MTA1)/NuRD complex [38,39]. By activating the Wnt signaling pathway, *ARMC8* has been linked to increased invasion in cutaneous squamous cell carcinoma and lung cancer [40,41,42]. Also implicated in the Wnt pathway, as well as the VEGF pathway, CEMIP (formerly known as KIA1199) has emerged via immunohistochemical studies as a possible biomarker for a variety of cancers [43,44,45]. The phosphatase INPP4B is an inhibitor of the Wnt pathway; its knockout has been linked to increased proliferation [46]. Knockout or downregulation of the transcription factor gene *FOXA1* has also been linked to worse prognosis, altering the carcinogenic activity of the Snail/Twist1 axis in breast cancer as well as prostate cancer [47,48]. *TBX3* has been linked to breast and cervical cancer proliferation, but it also inhibits the activity of the YAP/TAZ signaling pathway involved with cellular regeneration and growth [49,50,51].

Of note, *PTPN6*, a tyrosine phosphatase, affects cell growth and carcinogenesis in both bladder and colon cancer. Interestingly, overexpression of *PTPN6* has been associated with worse prognosis and increased metastasis, while the opposite was seen in other studies [52,53]. Additionally, there were cases where our findings were incongruent with other cancers. *PEBP4* expression was correlated with metastasis in colorectal and breast cancer but is significantly associated with better outcomes in our analysis. *AGER* (advanced glycosylation end-product-specific receptor) was shown to be associated with increased cell migration in cervical cancer, but the opposite as per our analysis [54]. The pro-carcinogenic effect that *ART1* has in colorectal cancer was not seen in our Bladder TCGA analysis [55].

Interestingly enough, some of the genes shown to be differentially expressed in regard to survival were not differentially expressed when comparing normal tissue to tumor; nor did their expression correlate to pathologic staging. For the pathologic staging, only *FN1*, *NRP2*, *FOXA1*, *NES*, *AGER*, *RUNX2*, *PTPN6*, *FBP1*, *TBX3*, and *STIM2* were shown to be significantly different.

While there may be differences in Stage I expression versus the other stages, our sample size (*n* = 2) was too small to reliably detect any except for in the expression levels of *FOXA1*. Instead, many of the differences were seen between Stage 2 versus 3 and 4. Histopathologically, this correlates with the expansion of the cancer through the muscle with possible lymph node and systemic metastasis. During this process, the pro-mobility changes which accompany the EMT would be fully evident. Our findings were mostly consistent with the survival analysis results, such as with *RUNX2* which showed worse prognosis with higher levels, increasing expression from Stages 1 to 4.

Through the network analysis, we were able to see more genes through which the EMT-related genes can modulate and affect carcinogenesis. For example, *VEGFA* was indicated, a well-known member of a family of growth factors which has proliferative and anti-apoptotic effects [56]. It has been found to be highly expressed in hepatocellular carcinoma (HCC) and triple-negative breast cancer (TNBC). It is associated with worse prognosis in HCC [57] and significantly lower progression-free survival in TNBC treated with chemotherapy [58]. Additionally, *SFRP2* was implicated, and recent immunohistochemistry work has linked this gene as a possible early biomarker of pancreatic and colon cancer [59,60]. Another gene highlighted in our analysis, *CAV1*, codes for a scaffolding protein and often predicts poor prognosis [61].

Enrichment analysis of the EMT-related genes further highlights the multiple modalities through which the EMT process itself encourages cancer growth and metastasis. Included among the highlighted cellular processes in our analysis are those regarding the establishment or maintenance of epithelial cell apical/basal polarity and positive regulation of cell migration. The VEGFA-VEGFR2 signaling pathway and associated angiogenesis, among the seminal hallmarks of cancer, were highlighted as the most overly enriched biologic process (*p* < 5.0 × 10^−7^). Other biological processes less typically associated with EMT such as “defense response to bacterium” and “signaling by interleukins” were picked up by the analysis. However, the same cytokine and inflammatory-mediated response to bacteria can help explain another way EMT-related genes play a role in cancer progression. The increased angiogenic and metabolic changes associated with certain proinflammatory states, which would help fend off a bacterial infection, can serve as fertile ground for initiation of cancer metastasis.

Among the most salient advances in our understanding of cancer progression is the complex role the immune system plays. Notable mediators of the immune system repeatedly occurred in our analysis. TGFBR1, a receptor for the TBFB growth factor, and CSF1R, a cytokine receptor, have been linked to pro-survival activity [62,63,64,65]. In other cancers, the chemokine CCL15 has been linked to tumor-associated macrophage recruitment and CCL18 has been linked to an immunosuppressive tumor microenvironment, allowing for evasion of the host’s immune system [66,67,68,69]. The most common immune cell type with a significant correlation was CD4+ T cells. While classically thought of as having an anti-tumor role, the varied subtypes not completely covered in our current deconvolution methods may play a paradoxical pro-tumor role. It is possible they do so by decreasing activation of other immune cells such as macrophages, warranting further study into the specific gene to immune cell interaction [70].

While this research represents a step forward in investigating the expression patterns of EMT-related genes, there exist certain limitations to these discoveries. These assessments rely on the levels of mRNA in a stable state as a proxy for protein levels. To partially address this, proteomic analysis of the highlighted genes in our study were examined in bladder cancer models through the depmap.org portal (https://depmap.org/portal/ (accessed on 12 September 2023)) and Cancer Cell Line Encyclopedia (CCLE), verifying expression of highlighted genes. Moreover, the functional behavior of these mRNA molecules could be subject to additional regulation by post-translational elements. Additionally, the techniques used to decipher immune cell infiltrations might be influenced by the tumor’s purity, contingent upon the specific formula employed. Ongoing efforts are dedicated to refining methods for accurately estimating cellular infiltrations based on mRNA levels. Overall, this study potentially serves as a starting point for future investigations aimed at directly scrutinizing the highlighted relationship between gene expression and prognosis, extending to the protein or enzymatic realm.

## Figures and Tables

**Figure 1 genes-14-01899-f001:**
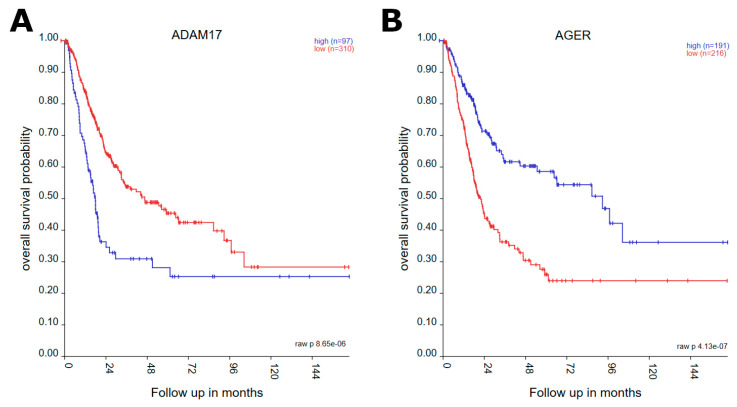
Subset of Kaplan–Meier plots based on mRNA expression of EMT-related genes which showed to be differentially expressed in relation to overall survival. Each plot lists the numbers of pts in “high” versus “low” mRNA expression cohorts based on the KaplanScan grouping algorithm. (**A**,**B**) show subset of Kaplan-Meier plots, based on gene expression levels of *ADAM17* and *ADER* respectively. See Appendix A for Kaplan–Meier plots of all genes shown to be differentially expressed.

**Figure 2 genes-14-01899-f002:**
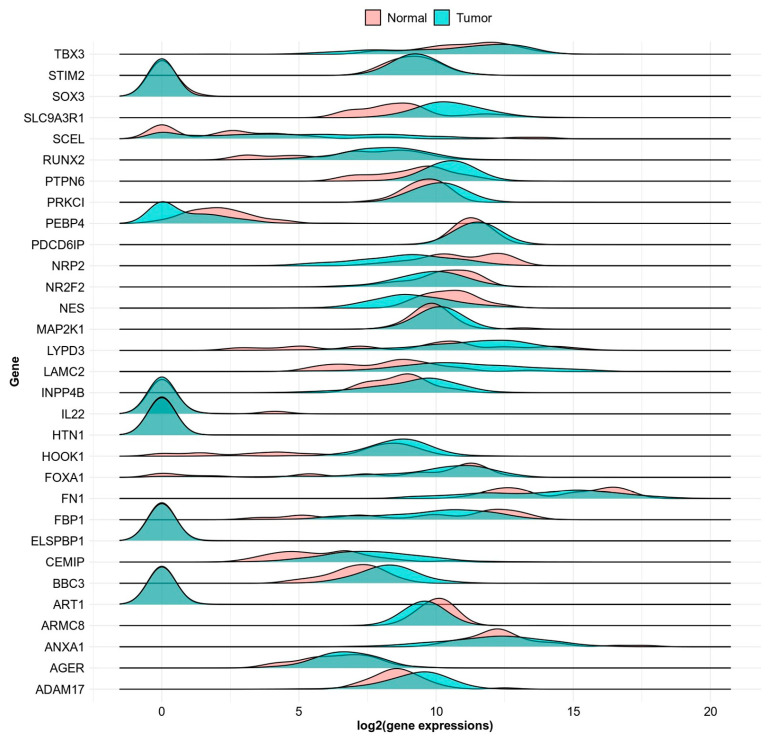
Density plot showing expression of normal versus tumor expressions for all selected genes which had both TCGA and GTEX data.

**Figure 3 genes-14-01899-f003:**
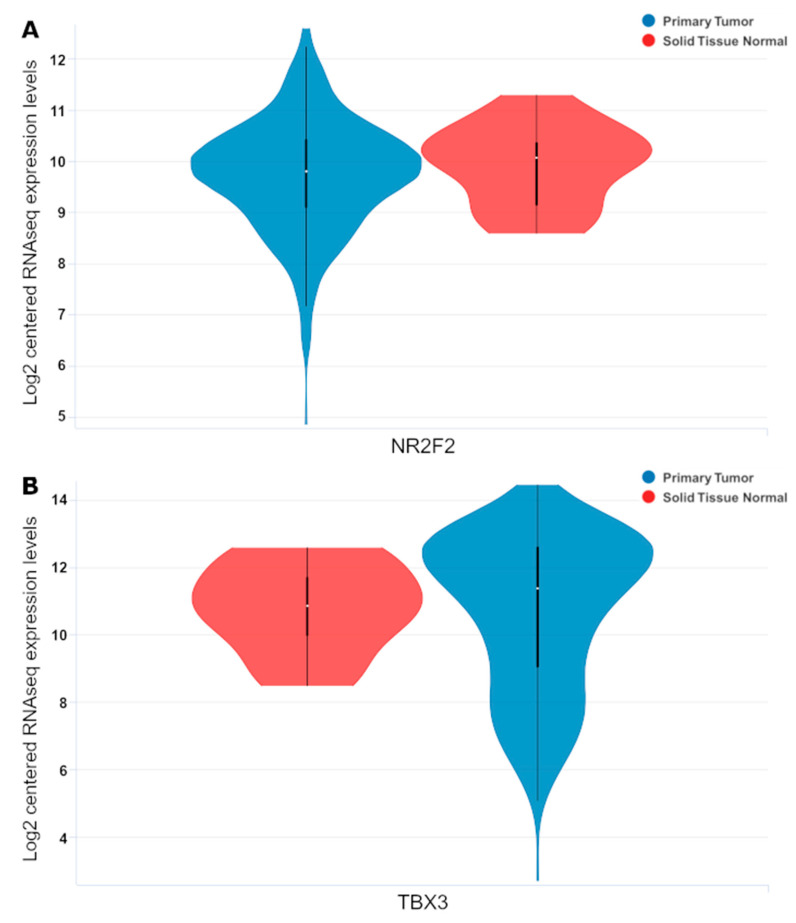
Subset of violin plot showing mRNA expression between primary tumors (Blue, *n* = 407) and surrounding normal tissue (Red, *n* = 19). For full listing, see Appendix A. (**A**,**B**) show subset of violin plots, showing primary versus normal solid tissue violin plots for *NR2F2* and *TBX3* respectively.

**Figure 4 genes-14-01899-f004:**
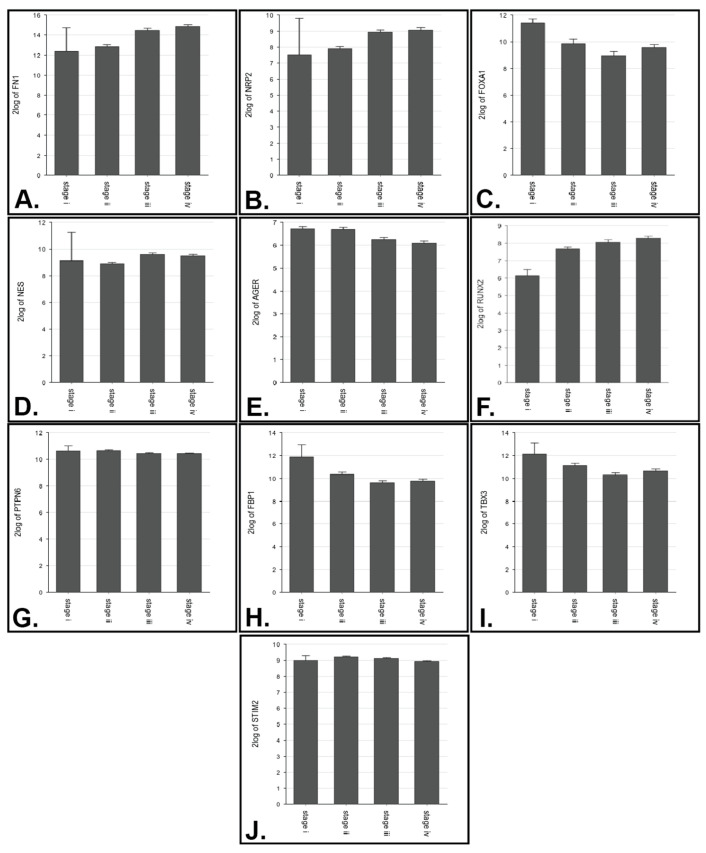
Bar plots showing log2 median mRNA expression levels of the EMT-survival related genes which also showed significant differences between any of the stages (Stage 1 *n* = 2, Stage 2 *n* = 130, Stage 3 *n* = 140, Stage 4 *n* = 134). (**A**–**J**) show genes for which expression levels were significantly different (stage plots of *FN1*, *NRP2*, *FOXA1*, *NES*, *AGER*, *RUNX2*, *PTPN6*, *FBP1*, *TBX3*, and *STIM 2* respectively).

**Figure 5 genes-14-01899-f005:**
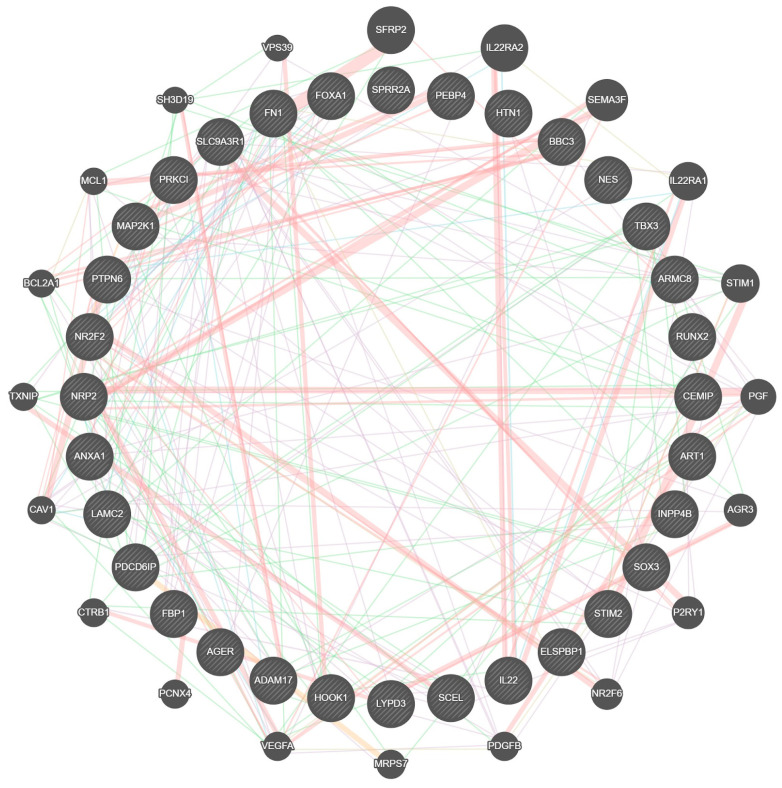
Network analysis showing 20 of the most highlighted genes. The analysis was conducted utilizing factors such as physical and anticipated interactions, protein co-localization, and shared DNA domains, alongside various other attributes. Connections between genes based on physical interactions are highlighted in red, shared pathways in blue, shared protein domains in yellow, predicted in orange, co-expression in purple, genetic interactions in green, and co-localization in navy.

**Figure 6 genes-14-01899-f006:**
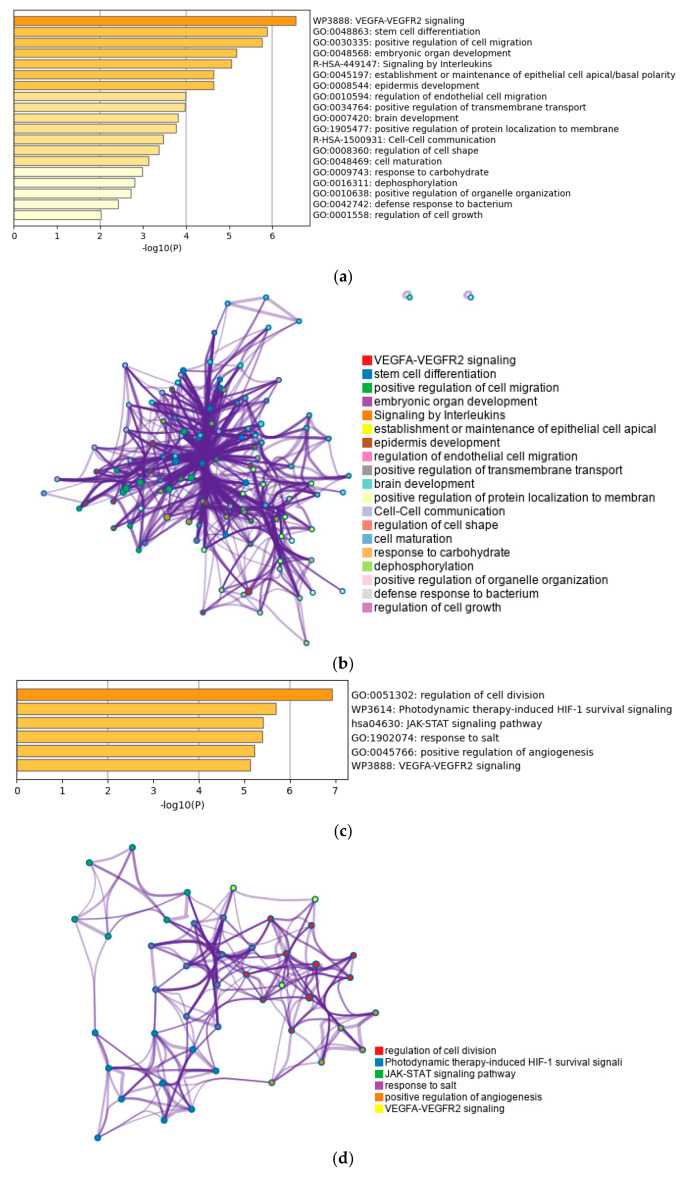
Enrichment analysis of the EMT genes involved in survival (**a**) as well as enrichment analysis highlighted in the aforementioned network analysis (**b**). Labeled are statistically enriched terms which are biologic pathways selected from KEGG and other hallmark gene sets. Additionally, for both the EMT genes and network highlighted gene ((**c**) and (**d**), respectively), the representative terms were converted into a network layout with each circle representing a single biologic process, grouped into larger “themes” as labeled in the color key. The size of the circle represents the amount of analyzed genes within that term.

**Figure 7 genes-14-01899-f007:**
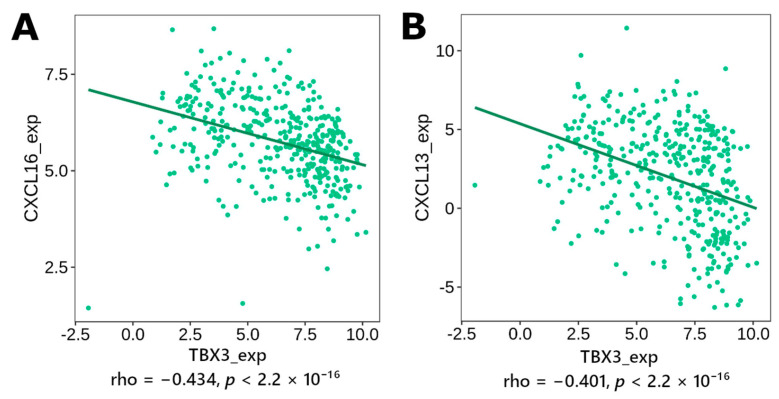
Sample Spearman’s correlation graphs for *TBX3* (full table of all EMT-related genes in Appendix B, Table A2), grouped by EMT-gene of interest. Only correlations shown to be statistically significant (*p* < 0.05 post multiple hypothesis correction) and deemed clinically significant |rho| > 0.4 are depicted both in the figure and included in the summary below. Subset shown, with (**A**) showing *TBX3* expression correlation with *CXCL16* expression and (**B**) showing correlation between *TBX3* and *CXCL13* expression; for other genes *NRP2*, *FN1*, *FOXA1*, *FBP1*, *ANXA1*, *LAMC2*, *HOOK1*, *NES*, *RUNX2*, and *PTPN6* see Appendix A.

**Figure 8 genes-14-01899-f008:**
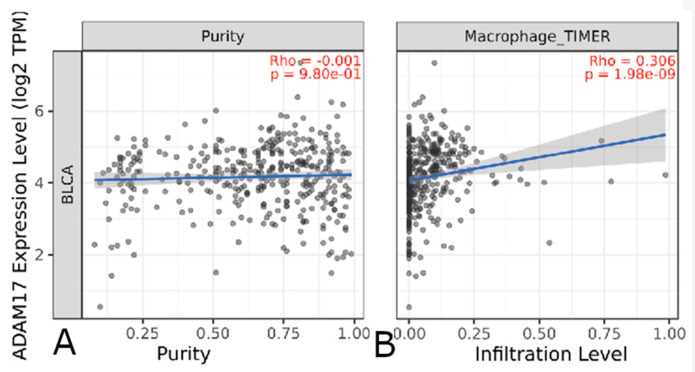
Correlations of various gene expression levels to tumor purity and immune infiltration. Only correlations to immune infiltration shown to be statistically significant (*p* < 0.05 post multiple hypothesis correction) and deemed clinically significant |rho| > 0.3 are depicted both in the figure and included in the summary below. Subset shown, for other genes *ANXA1*, *ARMC8*, *FBP1*, *FN1*, *FOXA1*, *LAMC2*, *MAP2K1*, *NRP2*, *PTPN6*, *RUNX2*, *STIM2*, *TBX3* see Appendix A. (**A**) shows correlation between *ADAM17* gene expression and tumor purity (amount of non-cancerous cells in tumor sample) and (**B**) shows correlation between *ADAM17* expression level and calculated macrophage infiltration.

**Table 1 genes-14-01899-t001:** Tabulated data from Kaplan–Meier plots based on mRNA expression of EMT-related genes which showed to be differentially expressed in regard to overall survival.

Gene	*p*-Value	Expression in Worse Prognosis
*ADAM17*	8.65 × 10^−6^	low
*AGER*	4.13 × 10^−7^	high
*ANXA1*	1.80 × 10^−6^	low
*ARMC8*	3.99 × 10^−8^	low
*ART1*	1.36 × 10^−3^	high
*BBC3*	5.28 × 10^−6^	high
*CEMIP*	1.83 × 10^−5^	low
*ELSPBP1*	1.65 × 10^−3^	high
*FBP1*	3.43 × 10^−5^	high
*FN1*	1.11 × 10^−5^	low
*FOXA1*	8.51 × 10^−5^	high
*HOOK1*	4.71 × 10^−12^	high
*HTN1*	1.02 × 10^−3^	high
*IL22*	4.18 × 10^−4^	high
*INPP4B*	2.37 × 10^−5^	high
*LAMC2*	1.98 × 10^−9^	low
*LYPD3*	3.33 × 10^−5^	low
*MAP2K1*	3.73 × 10^−7^	low
*NES*	2.91 × 10^−9^	low
*NR2F2*	1.34 × 10^−6^	low
*NRP2*	5.85 × 10^−5^	low
*PDCD6IP*	1.31 × 10^−5^	low
*PEBP4*	2.16 × 10^−4^	high
*PRKCI*	2.65 × 10^−4^	low
*PTPN6*	2.03 × 10^−7^	high
*RUNX2*	7.19 × 10^−8^	low
*SCEL*	2.27 × 10^−7^	low
*SLC9A3R1*	3.78 × 10^−4^	low
*SOX3*	3.92 × 10^−4^	high
*SPRR2A*	1.17 × 10^−5^	low
*STIM2*	2.11 × 10^−11^	high
*TBX3*	2.50 × 10^−4^	high

**Table 2 genes-14-01899-t002:** List of genes and *p*-values for which high mRNA expression is correlated with worse prognosis.

Gene	*p*-Value
*ADAM17*	0.0182
*ANXA1*	0.0433
*ARMC8*	0.0009
*CEMIP*	0.0300
*FN1*	0.0443
*LAMC2*	0.0050
*LYPD3*	0.0461
*MAP2K1*	0.0419
*NES*	0.0413
*NR2F2*	0.0495
*NRP2*	0.0481
*PDCD6IP*	0.0311
*PRKCI*	0.0457
*RUNX2*	0.0007
*SCEL*	0.0018
*SLC9A3R1*	0.0467
*SPRR2A*	0.0342

**Table 3 genes-14-01899-t003:** List of genes and *p*-values for which low mRNA expression is correlated with worse prognosis.

Gene	*p*-Value
*AGER*	0.0441
*ART1*	0.0446
*BBC3*	0.0317
*ELSPBP1*	0.0486
*FBP1*	0.0359
*FOXA1*	0.0356
*HOOK1*	0.0011
*HTN1*	0.0478
*IL22*	0.0389
*INPP4B*	0.0314
*PEBP4*	0.0322
*PTPN6*	0.0137
*SOX3*	0.0324
*STIM2*	0.0500
*TBX3*	0.0482

**Table 4 genes-14-01899-t004:** Violin plot genes and *p*-values.

Gene	*p*-Value	Higher Expression
*ADAM17*	8.31 × 10^−4^	Primary tumor
*AGER*	2.77 × 10^−2^	Primary tumor
*ANXA1*	1.41 × 10^−2^	Normal tissue
*ARMC8*	2.35 × 10^−4^	Normal tissue
*ART1*	1.73 × 10^−1^	Normal tissue
*BBC3*	5.39 × 10^−5^	Primary tumor
*CEMIP*/*KIAA1199*	1.93 × 10^−3^	Primary tumor
*ELSPBP1*	2.52 × 10^−1^	Normal tissue
*FBP1*	7.76 × 10^−1^	Normal tissue
*FN1*	3.88 × 10^−1^	Primary tumor
*FOXA1*	1.64 × 10^−1^	Primary tumor
*HOOK1*	5.13 × 10^−3^	Primary tumor
*IL22*	9.30 × 10^−2^	Normal tissue
*INPP4B*	2.06 × 10^−2^	Primary tumor
*LAMC2*	1.05 × 10^−4^	Primary tumor
*LYPD3*	3.03 × 10^−1^	Primary tumor
*MAP2K1*	2.88 × 10^−3^	Primary tumor
*NES*	8.36 × 10^−8^	Normal tissue
*NR2F2*	3.04 × 10^−1^	Normal tissue
*NRP2*	1.24 × 10^−4^	Normal tissue
*PDCD6IP*	3.67 × 10^−1^	Primary tumor
*PEBP4*	2.17 × 10^−4^	Normal tissue
*PRKCI*	4.31 × 10^−3^	Primary tumor
*PTPN6*	3.14 × 10^−3^	Primary tumor
*RUNX2*	1.90 × 10^−2^	Primary tumor
*SCEL*	2.05 × 10^−1^	Primary tumor
*SLC9A3R1*	8.95 × 10^−3^	Primary tumor
*SOX3*	3.35 × 10^−2^	Normal tissue
*SPRR2A*	2.77 × 10^−1^	Primary tumor
*STIM2*	8.06 × 10^−1^	Normal tissue
*TBX3*	7.13 × 10^−1^	Primary tumor

**Table 5 genes-14-01899-t005:** List of genes differentially expressed regarding overall survival when comparing normal versus tumor samples with *p*-values.

Gene	*p*-Value
*ADAM17*	0.0008
*ARMC8*	0.0002
*BBC3*	0.0001
*CEMIP*	0.0019
*LAMC2*	0.0001
*MAP2K1*	0.0029
*NES*	<0.0001
*NRP2*	0.0001
*PEBP4*	0.0002
*PTPN6*	0.0031

**Table 6 genes-14-01899-t006:** List of genes that approach significance after FDR correction with *p*-values (cutoff *p* value is 0.003143).

Gene	*p*-Value
*HOOK1*	0.0051
*HTN1*	0.0069
*SLC9A3R1*	0.0090

**Table 7 genes-14-01899-t007:** List of stages that showed significantly different expression of the relevant gene based off TCGA datasets.

Gene	Different Stages
*FN1*	2 vs. 3 (*p* = 1.22 × 10^−8^), 2 vs. 4 (4.20 × 10^−12^)
*NRP2*	2 vs. 4 (*p* = 2.10 × 10^−8^)
*FOXA1*	1 vs. 3 (*p* = 5.08 × 10^−3^)
*NES*	2 vs. 3 (*p* = 1.71 × 10^−5^)
*AGER*	2 vs. 3 (*p* = 2.00 × 10^−3^), 2 vs. 4 (*p* = 2.39 × 10^−5^)
*RUNX2*	2 vs. 4 (*p* = 2.99 × 10^−4^)
*PTPN6*	2 vs. 4 (*p* = 7.24 × 10^−3^)
*FBP1*	2 vs. 3 (*p* = 5.65 × 10^−3^)
*TBX3*	2 vs. 3 (*p* = 3.56 × 10^−3^)
*STIM2*	2 vs. 4 (*p* = 3.97 × 10^−5^)

**Table 8 genes-14-01899-t008:** Summary of Spearman’s correlation data between EMT-related gene and various immunomodulators, grouped by EMT-gene of interest.

Gene	Immunomodulator (Positive Correlation)	Immunomodulator (Negative Correlation)
*TBX3*		*CXCL16*, *CXCL13*, *CXCL11*, *CXCL10*, *CXCL9*, *CXCL5*, *CXCL3*, *CXCL2*, *CXCL1*, *CCL26*, *CCL23*, *CCL18*, *CCL13*, *CCL8*, *CCL7*, *CCL5*, *CCL4*, *CCL3*, *TIGIT*, *TGFBR1*, *PDCD1LG2*, *PDCD1*, *LAG3*, *IL10*, *IDO1*, *HAVCR2*, *CTLA4*, *CSF1R*, *CD274*
*NRP2*	*CXCL13*, *CXCL12*, *CXCL11*, *CXCL10*, *CXCL9*, *CXCL2*, *CCL26*, *CCL23*, *CCL21*, *CCL19*, *CCL18*, *CCL13*, *CCL11*, *CCL8*, *CCL7*, *CCL5*, *CCL4*, *CCL3*, *CCL2*, *LAG3*, *TIGIT*, *TGFBR1*, *PDCD1LG2*, *PDCD1*, *IL10*, *HAVCR2*, *CTLA4*, *CSF1R*, *BTLA*, *ADORA2A*	
*FN1*	*TGFBR1*, *TGFB1*, *PDCD1LG2*, *LAG3*, *IL10*, *HAVCR2*, *CSF1R*, *CD274*, *CXCL13*, *CXCL12*, *CXCL11*, *CXCL10*, *CXCL9*, *CXCL5*, *CXCL2*, *CCL26*, *CCL23*, *CCL21*, *CCL18*, *CCL13*, *CCL11*, *CCL7*, *CCL5*, *CCL4*, *CCL3*, *CCL2*	
*FOXA1*	CCL15	*CXCL12*, *CXCL11*, *CXCL10*, *CXCL9*, *CXCL5*, *CXCL3*, *CXCL2*, *CCL26*, *CCL23*, *CCL21*, *CCL18*, *CCL13*, *CCL8*, *CCL7*, *CCL5*, *CCL4*, *CCL3*, *CCL2*, *TGFBR1*, *TGFB1*, *PDCD1LG2*, *LAG3*, *IL10*, *HAVCR2*, *CTLA4*, *CSF1R*, *CD274*
*FBP1*	*CCL15*	*CCL4*, *TGFBR1*, *PDCD1LG2*, *CD274*
*ANXA1*	*PDCD1LG2*, *CD274*, *CCL7*	
*SPRR2A*	*CXCL8*, *TGFB1*	
*LAMC2*	*CXCL8*, *CXCL1*, *TGFB1*	
*HOOK1*	*TGFB1*, *CSF1R*, *CCL23*	
*NES*	*CXCL12*, *KDR*	
*PTPN6*	*LGAGLS9*	
*RUNX2*	*PDCD1LG2*	

**Table 9 genes-14-01899-t009:** Summary of Spearman’s correlation data between EMT-related gene and various immune cell types, grouped by EMT-gene of interest.

Gene	Immune Infiltrates (Positive Correlation)	Immune Infiltrates (Negative Correlation)
*ADAM17*	Macrophage, Neutrophil	
*AGER*		T cell CD8+
*ANX1*	T cell CD8+, Neutrophil, Myeloid dendritic cell	
*ARMC8*	Macrophage	
*FBP1*		T cell CD8+, Neutrophil, Myeloid dendritic cell
*FN1*	T cell CD4+, T cell CD8+, Macrophage, Myeloid dendritic cell	
*FOXA1*	T cell CD4+	T cell CD8+, Myeloid dendritic cell
*LAMC2*	T cell CD8+, Neutrophil, Myeloid dendritic cell	
*MAP2K1*	T cell CD8+, Neutrophil	
*NRP2*	T cell CD8+, Macrophage, Myeloid dendritic cell	
*PTPN6*		B cell
*RUNX2*	T cell CD8+, Myeloid dendritic cell	
*STIM2*	T cell CD4+	
*TBX3*		T cell CD8+, T cell CD4+, Neutrophil, Myeloid dendritic cell

## Data Availability

Publicly available datasets were analyzed in this study. This data can be found through https://www.cancer.gov/ccg/research/genome-sequencing/tcga (accessed on 1 September 2022) or on the R2 platform as described above.

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
