# Peer review of "Survival and Enrichment Analysis of Epithelial–Mesenchymal Transition Genes in Bladder Urothelial Carcinoma"

_genes, 2023, doi:10.3390/genes14101899_

Round 1
Reviewer 1 Report
The study you've outlined appears to be a valuable contribution to the field of bladder urothelial cancer research, particularly in exploring the role of epithelial mesenchymal transition (EMT)-related genes in prognosis and therapy. Here are some comments and suggestions for the authors:
1. It needs to verify the expression of EMT-related genes in the Cancer Cell Line Encyclopedia (CCLE)
2. Ensure that the introduction section clearly explains the background and significance of bladder cancer, the role of EMT in carcinogenesis, and why it's important to study EMT-related genes in bladder urothelial cancer specifically. This will provide context for readers who may not be experts in the field.
The quality of the English language in the provided text is excellent. The text is well-structured and effectively conveys the research objectives, methods, and findings. The use of terminology related to cancer research and molecular biology is accurate and appropriate. Overall, the writing is clear, concise, and free from grammatical or linguistic issues.
Reviewer 2 Report
The manuscript of Ali et al. investigates differential gene expression in samples from the publicly available TCGA bladder urothelial cancer dataset. The authors investigate 1184 Epithelial-Mesenchymal Transition (EMT) and identify a subset of 23 genes overexpressed in the urothelial cancer samples and correlate them with phenotypical data. These differentially expressed genes belong, among others, to the immunomodulatory, VEGF signaling, and bacterium response pathways.
Major comments:
1. The identified overrepresented genes are often unspecific and overexpressed in the majority of cancers. In this sense, the outcome of the study was predictable. Please state clearly what is the novelty of the study.
2. 32 overexpressed genes have been identified. Have there been any underexpressed genes?
3. The manuscript is too long and presents often too much data, instead of focusing on the most important findings. The manuscript would be more readable as a short communication than the full-length article repeating the same findings in a few places.
Minor comments:
1. The abstract is very general and too wordy. I would avoid long sentences.
2. Introduction: I would add a short paragraph about gene expression in urothelial cancer and shorten the first, general part of the introduction.
3. Methods: How was the list of 1184 genes created? Literature review? Details needed.
4. Methods: Please describe shortly TCGA dataset, cohort and provide a link to the database
5. Methods: The UCSC Xena platform should be described more in detail, and provide link. I have the feeling that Network & Enrichment Analysis is described much more in detail than other parts of the MEthod section, although it is not the most relevant.
6. Results: It would be better to present differentially expressed genes in the table instead of listing genes and their p-values in the main text.
7. Figures: There are two small, too many illustrations in one Figure (e.g. Figure 1, 3, 7, 5), and therefore illegible.
8. Figures: Figure numeration (Figure 5 is after Figure 7)
9. Discussion is too long, it would be better to use tables instead.
10. Discussion: I'm not sure if the word "biomarker" fits the contact. What kind of biomarker is meant? Are these differentially expressed genes and proteins validated as biomarkers?
11. Editorials: too many decimal places in p-values; names of the genes should be written in italics; some parts of the manuscript are wordy or unclear (e.g. " Several genes have been documented to be active in other cancers"); "KM estimates"-abbreviations should be spelled out,
Round 2
Reviewer 2 Report
Dear authors,
Thank you for the corrections and a comprehensive response.
Still, I would suggest the following:
1) Figure 1,3, 7, 8- to further reduce the number of graphs. I understand that they may be justified from your view, but for a reader they are often illegible.
2) Figure 3- the description of axes is too small and not visible even if I zoom in
3) Table 6 What is the significance level (lower than 0.05)? All p-values are lower than 0.05.
4) In the results section difference between stages is presented. Please describe different stages in the Introduction shortly.
5) Please read the manuscript again and check if there are any parts that can be shortened.